# Study of Metabolic Adaptation of Red Yeasts to Waste Animal Fat Substrate

**DOI:** 10.3390/microorganisms7110578

**Published:** 2019-11-19

**Authors:** Martin Szotkowski, Dana Byrtusova, Andrea Haronikova, Marie Vysoka, Marek Rapta, Volha Shapaval, Ivana Marova

**Affiliations:** 1Faculty of Chemistry, Brno University of Technology, 612 00 Brno, Czech Republic; xcszotkowski@fch.vut.cz (M.S.); xcbyrtusova@fch.vut.cz (D.B.); haronikova@fch.vut.cz (A.H.); xcvysoka@fch.vut.cz (M.V.); xcrapta@fch.vut.cz (M.R.); 2Faculty of Science and Technology, Norwegian University of Life Sciences (NMBU), 1430 As, Norway; volha.shapaval@nmbu.no

**Keywords:** carotenogenic yeasts, lipids, carotenoids, lipase, biosurfactants, ubiquinone, ergosterol

## Abstract

Carotenogenic yeasts are non-conventional oleaginous microorganisms capable of utilizing various waste substrates. In this work, four red yeast strains (Rhodotorula, Cystofilobasidium, and Sporobolomyces sp.) were cultivated in media containing crude, emulsified, and enzymatically hydrolyzed animal waste fat, compared with glucose and glycerol, as single C-sources. Cell morphology (cryo-SEM (cryo-scanning electron microscopy), TEM (transmission electron microscopy)), production of biomass, lipase, biosurfactants, lipids (gas chromatography/flame ionization detection, GC/FID) carotenoids, ubiquinone, and ergosterol (high performance liquid chromatography, HPLC/PDA) in yeast cells was studied depending on the medium composition, the C source, and the carbon/nitrogen (C/N) ratio. All studied strains are able to utilize solid and processed fat. Biomass production at C/N = 13 was higher on emulsified/hydrolyzed fat than on glucose/glycerol. The production of lipids and lipidic metabolites was enhanced for several times on fat; the highest yields of carotenoids (24.8 mg/L) and lipids (54.5%/CDW (cell dry weight)) were found in *S. pararoseus*. Simultaneous induction of lipase and biosurfactants was observed on crude fat substrate. An increased C/N ratio (13–100) led to higher biomass production in fat media. The production of total lipids increased in all strains to C/N = 50. Oppositely, the production of carotenoids, ubiquinone, and ergosterol dramatically decreased with increased C/N in all strains. Compounds accumulated in stressed red yeasts have a great application potential and can be produced efficiently during the valorization of animal waste fat under the biorefinery concept.

## 1. Introduction

In Europe, approximately 16 million tonnes of animal fat by-products are processed annually by fat processors and renderers, with Germany, France, the UK, and Spain as the main processing countries [1]. According to the Animal By-Products Regulations (1069/2009, 142/2011), animal fat by-products are divided into three categories. Category 1 includes the lowest grade fat material, which is only permitted to be used for energy generation. Category 2 material comprises condemned slaughterhouse fat by-products and dead farm stock, which, in addition to energy generation, may be used for technical applications. Category 3 material is edible fat, which is exclusively derived from approved animals and is obtained from slaughtered animals [2]. However, the fat itself is not intended for human consumption and is considered as an animal by-product, and can be used in animal feed (TSE, Tannlegeforeningens Systematiske Etterutdanning) [1,3]. In Europe, about 26% (806,000 tons) of animal slaughter fat by-products are classified as category 3 fat materials, according to EU regulations [1]. In addition to category 3 fat, 57% (1.5 million tons) of the total animal fat production accounts for edible animal fat, i.e., fat approved for human consumption [2,3]. The total amount of animal fat is expected to increase continuously, since consumers increasingly focus on healthy and low saturated fat diets [4]. In addition, meat production is expected to grow [2].

Upgrading the fatty acid composition of category 3 fat and the upgrading of edible rest fats thus has a great potential for valorization. The final ratio of unsaturated and saturated fatty acids (u/s) in the animal feed product is vital since it influences fat digestibility to a large extent [2]. Unfortunately, it is not economically sustainable to maintain a minimum u/s ratio in the final animal feed by the inclusion of unsaturated fat originating from other sources. Thus, sustainable processes for upgrading category 3 animal fat into high-value fat, for use in animal and fish feed, are desired.

Technologies used for modifying lipids can be grouped into chemical and biological technologies. For changing triacylglycerol composition by rearranging fatty acids and upgrading fat to a higher degree of unsaturation, interesterification is the most popular approach. Interesterification is performed using alkoxide or enzymes (lipases) [5]. Lipase-mediated transesterification leads to precisely structured triacylglycerols, which can be used in feed and food production processes. This process does not require expensive equipment and produces less waste than alkoxide-based processes. Nevertheless, the commercial exploitation of lipase-mediated upgradation of animal fats is limited, since the production of lipases is very costly [5].

Fermentation by oleaginous microorganisms is a biological technology for converting hydrophobic substrates, such as vegetable oils, volatile fatty acids. and animal fat. into higher value single cell oils that are rich in omega-3 and omega-6 fatty acids. Certain fungi, referred to as oleaginous, have the ability to accumulate up to 85% (*w*/*w*) lipid as a storage compound of the biomass [6]. In addition, some oleaginous fungi are able to perform simultaneous production of lipids and carotenoids [7]. The oleaginous fungi primarily accumulate lipids as triacylglycerols (TAGs), which are generally considered to be storage lipids. Fatty acids of TAGs are very similar to fatty acids in plant oils, where saturated and monounsaturated fatty acids dominate, and some ω6- polyunsaturated fatty acids (PUFAs) are present. Fatty acids derived from fungal single cell oils range from high-volume/low price to low-volume/high price fatty acids. Examples for high-volume/low price fatty acids are monounsaturated fatty acids and saturated fatty acids, used as surfactants, soaps, resins, stabilizers, etc., while low-volume/high price PUFAs (ω3-PUFAs) may achieve a market value of up to $150 US per kg in the pharmacy and food industries. Currently, fungi are in industrial use for the production of healthy PUFAs [8].

The kingdom Fungi (mycota) is a large group of eukaryotic organisms, which includes microorganisms such as yeasts and molds. Fungi have a worldwide distribution and grow in a wide range of habitats, including extreme environments. Fungi can degrade a wide range of simple and complex substrates, including rest materials, while producing a set of valuable metabolites [9]. So far, research and development of processes has focused on the utilization of different saccharide substrates [5,8], while the use of hydrophobic fat-based substrates has only recently attracted attention. Therefore, little is known about the use of different fat materials for the production of high-value fungal lipids [10,11,12].

Some yeast species accumulate carotenoid pigments, such as β-carotene, torulene, and torularhodin, which cause their yellow, orange and red colors, and are therefore called red yeasts [13]. Carotenogenic yeasts are a diverse group of unrelated organisms (mostly *Basidiomycota*) and the majority of the known species are distributed in four taxonomic groups [14]. The composition and amount of carotenoid pigments in numerous natural isolates of the genera *Rhodotorula*/*Rhodosporium* and *Sporobolomyces*/*Sporidiobolus* have been studied in detail [15].

During recent years, oleaginous fungi have been extensively studied for the production of low- and high-value lipids from different agricultural and food rest materials [9,10,11]. They are able to utilize many hydrophobic substrates, such as wastes of vegetable oil production, volatile fatty acids, and animal rest materials to co-produce other valuable chemicals, in addition to lipids [8,9]. Wastes such as palm oil mill effluent, crude glycerol [16], or even an industrial derivate of animal fat (stearin) [9], have been used for microbial lipid production by *Y. lipolytica*. Pork lard has been utilized for this purpose as well [17]. Nevertheless, there is extremely limited number of studies regarding the valorization of crude animal fat by yeasts other than *Yarrowia*. With the exception of our previous screening study [18], to our best knowledge, this is the first detailed report about the use of crude and processed (enzymatically hydrolyzed, emulsified) animal waste fat as the nutrition source for oleaginous red yeasts for the production of carotenoid/PUFA (polyunsaturated fatty acids)-enriched yeast biomass for feeding purposes. The described metabolites are produced in a higher degree during the assimilation of solid fat substrate by red yeasts. While some of these single cell factories, such as *Rhodotorula* spp., were evaluated as GRAS (generally recognized as safe), the production of red yeast biomass enriched in lipids, glucans, carotenoids, sterols, and ubiquinone, while producing lipase and biosurfactants, also seems to be a promising biotechnological strategy for the biorefinery concept [8].

Thus, our aim was to evaluate the ability of selected strains of carotenogenic yeasts to assimilate waste animal fat as a sole carbon source and to produce carotenoids, microbial oils, and other added value products, in order to assess the potential utilization of some of red yeasts as a biorefinery platform. The effects of selected parameters (pH, C/N ratio, enzyme hydrolysis, the addition of emulsifiers) on the growth and metabolic activity of selected red yeast strains were assessed.

## 2. Materials and Methods

### 2.1. Materials

Fresh mixed animal fat was provided by NorskProtein Co. (Norway) and kept at −20 °C until further use. Fat processing was done according to Reference [18], using mixed samples from different animal sources. The storage time was no longer than 2 weeks. Antioxidant Thermox liquid FG (Food Grade) was added to the fat. The fatty acid composition of the animal fat was verified by NorskProtein. About 80% of the fatty acids were composed from the sum of palmitic acid (23.3%), stearic acid (18.4%), and oleic acid (39.1%), while the summary content of linoleic and linolenic acids represented about 7%.

Sunflower oil for the oil spreading test was purchased from market retail. Yeast extract, peptone, yeast nitrogen base w/o, Tween 80, ammonium sulfate, magnesium sulfate (heptahydrate), glucose, glycerol, sulfuric acid, sodium hydroxide, potassium dihydrogenphosphate, anthracene, the Lipase Basic Kit, and lipase from *Candida rugosa* (500 mg/L) were purchased from Sigma-Aldrich (Missouri, USA).

### 2.2. Yeast Strains

Based on our preliminary results [18], four red yeast strains were enrolled in the study, as follows: *Cystofilobasidium macerans* CCY (Culture Collection of Yeasts) 10-1-2, *Rhodotorula glutinis* CCY 20-2-26, *Rhodotorula mucilaginosa* CCY 19-4-6, and *Sporobolomyces pararoseus* CCY 19-9-6. All strains were purchased from Culture Collection of Yeasts (CCY; Institute of Chemistry, Slovak Academy of Sciences, Bratislava, Slovak Republic), preserved on malt agar, and stored at 4 °C in darkness.

### 2.3. Media Preparation and Cultivation of Yeasts

#### 2.3.1. Enzyme Hydrolysis of Animal Fat

Animal fat hydrolysis was carried out by selective triacylglycerol esterases. Glycerol and free fatty acids were formed as the final products of enzyme hydrolysis. First, a set of commercially available lipases from different sources (yeasts, fungi, bacteria, porcine—Sigma-Aldrich) were used for testing the hydrolysis degree (summarized in Table 1). The reaction mixture contained 5 g of animal fat and 25 mg of enzyme preparative in a total volume of 50 mL of phosphate buffer (pH 7.2). The specific activities of individual enzymes were similar, as introduced in Table 1. Enzyme hydrolysis was performed under permanent stirring (300 rpm) at 37 °C for a total period of 30 h in two repetitions. Samples were taken after 1, 3, 6, 24, and 30 h of hydrolysis for glycerol analysis (see Section 2.4.6). All of the enzyme hydrolysis experiments were performed in duplicate. After finishing hydrolysis, the degree of fat degradation was analyzed by TLC (Thin Layer Chromatography) (see Section 2.4.5). According to the results obtained in the pilot experiments, lipase from *Candida rugosa* was the most effective and, thus, this enzyme was used for the hydrolysis of animal fat in all further experiments.

Enzyme hydrolysate for media preparation was produced as follows. Crude animal fat was put into 0.1 mol/L phosphate buffer pH 7.2 in a ratio of 1:10 (*w*/*v*). The mixture was heated to 37 °C and consistently stirred. Lipase from *Candida rugosa* (specific activity 2 U/mg of protein) was added to the mixture in the amount of 5 mg/g of fat and incubated for 24 h. After the hydrolysis, the water phase with glycerol was separated from the upper lipid phase containing the rest of the fat and free fatty acids. The water phase was analyzed for glycerol content (see Section 2.4.6) and used as a media component.

#### 2.3.2. Media Composition

Yeast strains were cultivated using two-step inoculation. The cultivation of the inocula was performed in Erlenmeyer flasks. The first inoculum (50 mL) was cultivated for 24 h at 28 °C, under continuous lighting and shaking. Inoculum I (20 mL) was then transferred into 100 mL of fresh inoculum II (volume ration 1:5), which was grown under the same conditions as inoculum I. The composition of inoculation media was as follows: Glucose 20 g/L, peptone 20 g/L, yeast extract 10 g/L and salts—KH_2_PO_4_ 4 g/L, (NH_4_)_2_SO_4_ 4 g/L, and MgSO_4_·7 H_2_O 0.696 g/L.

Summary of cultivation of each of the four studied yeast strains in the production media:
(a)Cultivation in Erlenmeyer flasks (50 mL of basic medium) on glucose for the construction of growth curves in the time period 0–146 h/two parallel cultivations; evaluation of means (*n* = 2)(b)Cultivation in Erlenmeyer flasks (50 mL of medium) in 6 different media (C sources: Glucose, glycerol, crude fat, emulsified fat, hydrolyzed fat, and crude fat + enzyme; C/N ratio = 13; see below) for 96 h for (i) continuous measurement of biomass, pH of media, and lipase production in the time period from 0–96 h and (ii) continuous measurement of lipid and metabolite production during the time period from 60–96 h (supposed stationary phase)/three parallel cultivations; evaluation of means ± SD (*n* = 3)(c)Cultivation in Pyrex flasks (500 mL of medium) in four different media (C sources: Glucose, glycerol, crude fat, and emulsified fat) for 96 h, at 4 different C/N ratios in the range of 13–100, to measure the production of biomass and metabolites. Some of these media were evaluated for biosurfactant production)/three parallel cultivations; evaluation of means ± SD (*n* = 3)


For growth curves (a) and continuous measurement (b) of growth, the pH of the media, the biomass and the metabolite production, 96 h cultivation was performed in 250 mL flasks, each with 50 mL of medium. After 24 h, inoculum II was transferred into production media in Erlenmeyer flasks in the ratio of 1:5. The production media contained the same amount of salts as the inoculation media (KH_2_PO_4_ 4 g/L, (NH_4_)_2_SO_4_ 4 g/L and MgSO_4_·7 H_2_O 0.696 g/L.) and different types of carbon sources. Generally, four types of production media according to C source were used for yeast cultivation, as follows: (i) Basic media with a single C source—crude animal fat (17.8 g/L), glycerol (30 g/L), or glucose (30 g/L); (ii) the production medium containing animal fat (17.8 g/L) and an enzyme (500 U/L of enzyme from *Candida rugosa*); (iii) the production medium with crude animal fat (18.7 g/L) and an emulsifier (Tween 80, 5 mL/L); and, finally, (iv) the production medium with previously enzymatically hydrolyzed animal fat (glycerol concentration 30 g/L), all diluted in the buffer containing KH_2_PO_4_ 4 g/L, (NH_4_)_2_SO_4_ 2 g/L, and MgSO_4_·7 H_2_O 0.696 g/L, and with the pH adjusted to 5.5–6.0. The value of the carbon/nitrogen ratio was C/N = 13 in all these media. Cultivation was carried out at 28 °C, pH 5.5, under constant stirring in a laboratory shaker and under permanent light exposure, which is necessary for carotenoid production [18,19]. Samples were taken in regular 6 h intervals for the total period of 96 h of cultivation. Cells were collected by centrifugation (5000 rpm, 10 min) for analysis of the biomass and intracellular metabolites. In the supernatant, the extracellular lipase activity and the pH of the cultivation medium were analyzed.

Cultivation in media with different C/N ratios in the range of 13–100 (c) was performed in a larger volume and with more intensive aeration. In these experiments, inoculum II was transferred into 2 L cultivation flasks (Pyrex) containing 500 mL of the appropriate sterile production medium. Cultivation was carried out at 28 °C, pH 5.5, under constant stirring (800 rpm) and under permanent light exposure. Cultivations were done in triplicates. Media containing glucose, glycerol, animal fat, and emulsified animal fat as the C-source were prepared as follows. An appropriate amount of C-source (see Table 1) was added into 1000 mL of water with salts—KH_2_PO_4_ 4 g/L, (NH_4_)_2_SO_4_ 4 g/L, and MgSO_4_·7 H_2_O 0.696 g/L. As an N-source (NH_4_)_2_SO_4_ was used and the C/N value was determined as a ration of the carbon content in the C source and the nitrogen content in the N source. Due to the complicated evaluation of the C/N ratio, the hydrolyzed media were not used.

### 2.4. Analytical Methods

#### 2.4.1. Cell Dry Weight

Fermentation broth samples (10 mL) were centrifuged at 6000 rpm g for 2 min to spin down the yeasts. The supernatant was collected for further analyses (lipase, biosurfactants, pH) and stored at −20 °C. Then, the yeast cells were washed twice with a mixture of distilled water and hexane 1:1 (*v*/*v*) and suspended in 1 mL of distilled water. Then, the purified biomass was quantitively transferred into Eppendorf tubes, frozen at −20 °C, and then freeze-dried. After determining their weight, to calculate CDW, the dried cells were used for the analysis of carotenoids, ergosterol, ubiquinone, lipids, and glucans.

#### 2.4.2. Lipase Activity

The lipase activity was determined by colorimetry using p-nitrophenylpalmitate as a substrate. Lipase hydrolyzes this substrate to form yellow-colored p-nitrophenol, which can be determined at 405 nm. A calibration curve of p-nitrophenol was constructed in the range of 0.005 to 0.05 mmol/L [18]. The stock solution of substrate contained 0.0135 g o-nitrophenyl palmitate, 0.0170 g sodium dodecylsulphate, and 1 g of Triton X-100, dissolved in 100 mL of distilled water. This solution was stored no longer than 3 days at 4 °C.

The supernatant of the yeast cultivation was used as a sample for extracellular lipase determination. After centrifugation (6000 rpm, 2 min) the reaction mixture was prepared in a 96-well microtitration plate using 50 µL of supernatant, 150 µL of 0.1 M phosphate buffer, pH = 7.2, and 100 µL of substrate solution. The enzyme reaction was recorded by measuring the absorbance for 60 min at 405 nm in an ELISA reader (BioTec, SRN). Absorbance was measured in 5 min intervals and lipase activity was evaluated according to Equation (1), as follows:
(1)$$a=\frac{A_{405}+d}{c\cdot V\cdot t}$$
where *A*_405_ is the absorbance; *c* and *d* are the coefficients of the calibration regression curve *A* = *cx* + *d*; *V* is the supernatant volume; and *t* is the time period of measurement.

The lipase activity was expressed in international units (U). One unit was defined as the amount of enzyme which resulted in the formation of 1 nmol of p-nitrophenol during 1 min (per mL of medium).

#### 2.4.3. Biosurfactants

For preliminary qualitative analysis of the biosurfactants, an oil spreading test with a supernatant of cultivated yeasts was used. Cells were cultivated in glucose medium (see Section 2.3.2) for 96 h. Biosurfactant production was induced by sunflower oil. The test was performed in 90 mm Petri dishes, where the oil layer was applied. Nile red was used for a better visualization of the oil spreading process. Positive (Triton X-100) and negative (water) controls were included into the test.

For semiquantitative determination of biosurfactants, method based on the solubilization of crystalline anthracene was used. The supernatant (5 mL) and both positive and negative controls were mixed with 50 μL of hydrophobic anthracene and stirred at laboratory temperature for 24 h. After incubation, samples were centrifuged at 6000 rpm for 5 min. Then absorbance at 354 nm was measured in a quartz cuvette. This absorbance is related to the amount of anthracene dissolved in the surfactant present in the sample.

#### 2.4.4. Carotenoids

Carotenoids, ergosterol and ubiquinone, were quantified by the simultaneous high-performance liquid chromatography (HPLC/PDA (Photo-diode Array Detector)) method. After finishing, the cultivation samples were lyophilized (see Section 2.3.2) and 20 mg of lyophilized biomass was used for analysis. For rehydration, 50–100 µL of distilled water was added to the dried sample. The rehydrated biomass was then mixed with 5 times the amount of glass beads and mechanically disrupted. Further, the pellet was extracted twice by 1 mL of acetone using vortex homogenization for 15 min. After homogenization, the extracts were collected and dried on a rotary evaporator at 40 °C. Dry extract was dissolved in 1 mL of ethyl acetate and stored at −40 °C.

Before analysis, samples were filtered using 0.45 μm PTFE filters and 10 μL of each sample was injected onto a Kinetex C18 column, 5 μm, 150 × 4.6 mm (Phenomenex, USA), with a guard column, 5 μm, 30 × 4.6 mm, and both were equilibrated to 45 °C with a mixture of acetonitril:methanol:Tris HCl buffer, (84:2:14), pH = 8, as mobile phase A. Mobile phase B was composed of methanol:ethylacetate (60:40). Gradient elution according to Table 2 was used (flowrate 1 mL/min), on a Thermo Finnigan Surveyor HPLC/PDA system. Carotenoids were detected at 445 nm, while ubiquinone and ergosterol were detected at 285 nm. Xcalibur software was used for chromatography data analysis. The total carotenoids, ergosterol, and ubiquinone content were evaluated using external calibration, as in Reference [19].

#### 2.4.5. Lipids and Fatty Acids

The total lipids and individual fatty acids were determined by optimized gas chromatography/flame ionization detection (GC/FID) analysis. Lipids were extracted from 10 mg of freeze-dried cells by the Folch method (methanol:chloroform, 2:1) and converted into their fatty acid methyl esters (FAMEs). The FAMEs were analyzed by gas chromatography (GC) analysis. GC analysis of FAMEs was carried out on a TRACE^TM^ 1300 Gas Chromatograph (Thermo Fischer Scientific, USA) equipped with an Al 1310 autosampler, a flame ionization detector, and a Zebron ZB-FAME column, where the bleed specification at 260 °C was 3 pA (30 m, 0.25 mm, 0.20 μm). The temperature program is introduced in Table 3. Fatty acids were identified by comparison to commercial FAME standards (FAME32, Supelco) and quantified by the internal standard method, involving the addition of 100 μg of commercial dodecanoic acid (Sigma-Aldrich). Chromatography data were evaluated by Chromeleon software. The total lipid concentration, based on GC data, was evaluated as well.

The lipid fractions after the enzyme hydrolysis of animal fat by individual lipases (Table 1) were also analyzed qualitatively by the thin layer chromatography technique. A Silikagel plate with applied samples of enzymatically hydrolyzed animal fat was placed into chamber containing the elution mixture (hexane:diethylether:acetic acid, 70:30:1). After elution and drying, separated fractions were visualized by the mixture of 10% H_3_PO_4_ and 10% CuSO_4_ in distilled water at 110 °C for 10 min.

#### 2.4.6. Glycerol

Glycerol concentration in media with hydrolyzed fat was analyzed by high performance liquid chromatography (HPLC) with refractive index detection, according to Reference [18]. The HPLC/refractive index (RI) analysis of glycerol was carried out on a Dionex Ultimate 3000 (Thermo Fischer Scientific, USA) apparatus using a Rezex ROA Organic acids column (Phenomenex; 300 × 7.8 mm). The sample (20 μL) was applied into the column and equilibrated by an elution mixture (0.005 mol/L sulfuric acid) at 60 °C. Separation was performed under isocratic elution (flow 1.0 mL/min) for 15 min. Chromatography data were evaluated using the Chromeleon software.

#### 2.4.7. Glucans

The content of total glucans in freeze-dried yeast biomass was determined according to the Yeast and Mushroom β-glucan Assay Procedure K-YBGL (Megazyme Int., Poland). To estimate the total glucan content, the yeast biomass was hydrolyzed with ice-cold 12 M sulphuric acid for 2 h and then incubated for 2 h at 100 °C. Further, the neutralization of acidic hydrolysate with 200 mM sodium acetate buffer (pH 5) and 10 M KOH was completed, followed by the effect of enzymes exo-β-(1→3)-D-glucanase and β-glucosidase in acetate buffer (pH 4.5) for 1 h at 40 °C. The α-glucan content was determined after enzymatic hydrolysis with amyloglucosidase and invertase. The β-glucan content could be calculated from the assay kit procedure as the difference between the total glucan content and the α-glucan content. The absorbance values for the total glucan contents were obtained spectrophotometrically at 510 nm after adding a glucose oxidase/peroxidase reagent to the samples.

### 2.5. Electron Microscopy

For cryo-scanning electron microscopy (SEM) imaging, the samples of *Rhodotorula glutinis* CCY 20-2-26, *Cystofilobasidium macerans* CCY 10-1-2, and *Sporobolomyces pararoseus* CCY 19-9-6 were prepared as follows: (i) The yeast samples were cultivated aerobically at 25 °C in glucose medium at different C/N ratios for 144 h; (ii) cell suspensions were fixed in 2.5% glutaraldehyde in PBS (Poly-Buffer Solution) for 2 h and 30 min in 1% OsO4, dehydrated by ethanol series, and dried in HDMS (Hexamethyldisilazane) on the glass slides. Images of the prepared samples were scanned without any metal coating at an electron beam energy of 1 keV and a beam current of 6.3 pA in an SEM Magellan (FEI).

Samples of the yeast cultures were visualized using a Transmission Electron Microscopy JEOL 1010 following the protocol described in our previous reports [20,21].

### 2.6. Statistical Analysis

The growth experiments in Erlenmeyer flasks and in 2l Pyrex flasks were carried out in triplicate and duplicate, respectively (see Section 2.3.2). The presented results are the mean of the replicates, and the standard deviations are shown as error bars in the figures. Data handling and statistics were performed using the Excel software package (Microsoft Excel 2013, Microsoft Corp., Redmond, WA, USA).

## 3. Results

### 3.1. Animal Fat Processing

Yeasts are able to utilize fats in crude form. The addition of an emulsifier could increase the bioavailability of the solid waste fat to the yeast cells [17]. Another method is hydrolysis of the waste fat to glycerol and fatty acids or fatty acid salts. In the present study, enzyme hydrolysis by microbial lipase was performed. First, a set of commercially available lipolytic enzymes were tested to degrade animal fat (Table 4). The specific activity of tested enzymes was similar and ranged from 1 to 15 U/mg of protein. Based on the glycerol concentration formed during the hydrolysis of animal fat (Table 4), and to the degree of hydrolysis (Figure 1), the enzyme from *Candida rugosa* (Sigma-Aldrich) was evaluated as the best choice. After hydrolysis, no fraction of TAG was visualized, most of the fat was degraded to fatty acids and quite low fractions of DAG (diacyl glycerols) and MAG (monoacyl glycerols) were detected. The amount of glycerol obtained from the degradation of animal fat was the highest (Table 4), and, moreover, the rest of the partially degraded lipids could serve as emulsification agents. This enzyme was then used for the preparation of all media containing hydrolyzed fat or as an addition to crude fat. In media with hydrolyzed fat, the rest of external enzyme was inactivated by sterilization. In some experiments, crude fat substrates with the native enzyme directly added to the production media were used for comparison.

### 3.2. Growth Parameters and Biomass Production in Different Fat-Based Media

The aim of this study was to perform detailed research of the biomass production and the overall metabolic activity during the growth of four red yeast strains (*Cystofilobasidium macerans* CCY 10-1-2, *Rhodotorula glutinis* CCY 20-2-26, *Rhodotorula mucilaginosa* CCY 19-4-6, and *Sporobolomyces pararoseus* CCY 19-9-6). First, growth curves on glucose medium were constructed for 146 h. Two different C/N ratios were used, and selected examples were documented (Appendix A; *R. mucilaginosa* CCY 19-4-6 and *S. pararoseus* CCY 19-9-6). The shape of the growth curves of all the tested strains are characteristic for carotenogenic yeasts and indicate that after 50–70 h of growth the logarithmic phase was finished and followed by the typical prolonged stationary phase with some local maxima and minima of growth, as described previously (19). These fluctuations are probably caused by the gradual formation and utilization of storage compounds. In both strains, biomass production was slightly lower at C/N 50 when compared with C/N 13. After 90 h, a relatively well-balanced stationary phase was reached in both strains. This finding was used for the planning of further experiments, where the total period of all cultivations was 96 h. This could probably be the optimum phase for production of enough biomass with sufficient contents of valuable secondary metabolites.

Biomass production was strain specific depending on the type of medium (Figure 2). While growth on glucose and glycerol was relatively similar in all strains, larger differences between individual strains were caused by fat presence in the media. Similar to in the screening study, the total amount of biomass produced on glycerol was about 20% lower than on glucose. All fat-containing media were utilized by red yeasts to a similar or slightly higher degree than glycerol alone, with the exception of *S. pararoseus*, which produced several times higher biomass on fat-containing media (Figure 2c–e). All strains exhibited the best growth on hydrolyzed fat; about two times higher biomass was observed in *S. pararoseus* CCY 19-9-6 and *R. glutinis* CCY 20-2-26 when compared with glycerol and glucose. Growth curves recorded over 96 h were different according to the strain and media composition. In all strains, a decrease of growth velocity could be observed around the 50th hour of cultivation. While in *C. macerans* CCY 10-1-2 and *R. mucilaginosa* CCY 19-4-6, the shape was similar and a stationary phase was probably reached (see Appendix A). The other strains (*R. glutinis* CCY 20-2-26, *S. pararoseus* CCY 19-9-6) mainly exhibited less typical shapes in fat-containing media. In *S. pararoseus* CCY 19-9-6, the growth was strongly supported by the presence of fat in the medium. A cultivation period of 96 h was used as a compromise value based on previous experiences with red yeasts. A preliminary study showed a prolonged stationary phase with some local minima and maxima due to the periodic utilization of the endogenous substrate and the formation of storage compounds [18,19].

During growth, the pH of the medium was monitored in all cultivations. While growth on glucose led to a significant decrease of pH (Figure 3) in all strains (mainly *S. pararoseus* CCY 19-9-6 and *C. macerans* CCY 10-1-2), in the glycerol medium the pH was decreased only slightly during 20–30 h of growth and then the pH was increased to the starting value. A similar decrease in pH around 30th hour was also observed in all the fat-containing media except hydrolyzed fat, where the pH was unchanged during the whole growth. The decrease of pH is probably interconnected with the carbon flux of organic acid (mainly citric acid) production, which was the highest in the glucose medium [8,19].

### 3.3. Extracellular Lipase Production

Lipases have been found in many species of animals, plants, and microorganisms. Many microorganisms secrete lipases during growth on organic residues. Lipase production depends largely on microorganisms, culture practices, medium composition, and cultivation design [22].

In this work, the lipase production into the media was measured simultaneously with the biomass production and pH during the whole growth (Figure 4). The production of extracellular lipase was probably induced by presence of animal fat in the cultivation medium. Nevertheless, according to data introduced in Figure 4, it is clear that the production of lipase also occured to a low degree on glucose and the glycerol medium. On fat media, enzyme induction can be predominantly found in the strains R. mucilaginosa CCY 19-4-6 and C. macerans CCY 10-1-2, cultivated on crude fat (Figure 4c). In these strains about a 5× increase of lipase activity in production medium could be found at about the 48th h of growth when compared with T = 0. The induction of lipase in these strains is associated with the increased production of biomass and also with the total lipid production on crude fat (Appendix A). In C. macerans CCY 10-1-2 about w 3× higher production of lipids was observed on crude fat than on glycerol as a C source. Similarly, in S. pararoseus CCY 19-9-6, a 7× higher lipid production on crude fat was found when compared with the glycerol medium, respectively.

Extracellularly produced lipase by red yeasts exhibited activity in the range of 0.01 to 0.025 U/mL, which is about 20–50× lower than the activity of the commercial enzyme from *C. rugosa* used for external hydrolysis. The induction of extracellular lipase on hydrolyzed fat was lower than on crude fat (Figure 4d). This effect could be caused by some degradation products of lipids present in the medium. The medium with crude fat and the addition of the enzyme at the beginning of the production phase led to a similar induction of the lipase as in medium with crude fat only (Figure 4e). Thus, the presence of the extracellular enzyme at the activity of 0.5 U/mL exhibited no inhibition effect on the production of extracellular lipase. Oppositely, the addition of the emulsifier (Tween 80) fully inhibited lipase production. Based on data measured during growth (Figure 2 and Figure 3), the decrease in pH in the 30th hour of growth was also associated with some increase in lipase production, while enzyme induction is probably strain and medium specific. Generally, the activity of extracellular lipase in yeast media is relatively low, which can be connected with the high degree of dilution and probably with some degree of interaction of the lipase with the yeast cell surface [18,23].

### 3.4. Biosurfactant Production

Lipases and biosurfactants are compounds produced by the microorganisms thst are generally involved in the metabolization of oil substrates. However, the relationship between the production of lipases and biosurfactants has not been established yet. When tested yeast strains were cultivated on glucose, as well as on glycerol and crude fat, the diameter of the clear zone obtained by the oil spreading test method was more than 8 cm or the lipid layer was totally emulsified (see Section 2.4.3, Appendix A). According to oil spreading test, the emulgation activity decreased in the course of C. macerans CCY 10-1-2, S. pararoseus CCY 19-9-6, R. mucilaginosa CCY 19-4-6, and R. glutinis CCY 20-2-26. Using this qualitative test, the emulsification effect of yeast media is relatively high and comparable to commercial detergent (Tween 100) at a concentration of 6.5 g/L.

Approximate quantification of biosurfactant production was done by the method based on the solubilization of crystalline anthracene (see Section 2.4.3). The absorbance index values related to the amount of biosurfactant are introduced in Figure 5, where data for two different C/N ratios are shown. Biosurfactant production was studied in media with glucose, glycerol, and crude fat as the C-source. The ratio C/N 13 represents a medium with a relatively small amount of carbon substrate, while C/N 50 contains about a four times higher substrate content, e.g., solid waste fat. Data obtained in this experiment confirmed the above-mentioned results of the orientational oil spreading tests. The highest production of biosurfactants was detected in C. macerans CCY 10-1-2 and S. pararoseus CCY 19-9-6, a lower production was detected in R. mucilaginosa, and R.glutinis was found to be the worst producer in both tests. The production of biosurfactants at C/N 13 was several times higher than at C/N 50 in all tested strains and on all C-sources used, with the exception of C.macerans CCY 10-1-2 growing on crude fat (Figure 5). Simultaneous production of lipase and biosurfactants was observed in fat-based media, predominantly in the strains C.macerans CCY 10-1-2 and S.pararoseus CCY 19-9-6, which belong to the best producers of lipids on the crude fat substrate (Appendix A).

### 3.5. Metabolite Production in Fat-Based Media Rearranged

#### 3.5.1. Production of Carotenoids and Sterols during Growth in Fat Media

Production of lipids and other metabolites by tested yeast strains was studied during the early stationary phase of growth (60–96 h) in selected media at C/N 13 to complete the data introduced in Figure 1, Figure 2 and Figure 3 (see Section 2.3.2, b)). The values of total carotenoids, ergosterol, ubiquinone (mg/L of culture), and total lipids (% of cell dry mass) are means (*n* = 3). As a consequence of the illustrative character of these data, SDs are not introduced in the Appendix A. The values of SD were in the range of 2–10% for metabolites and 0–5% for biomass determination, similar to the data present in Appendix A.

In further experiments, emulsified fat and hydrolyzed fat were used as C-sources according to Section 2.3.2, b). Cultivation was done for 96 *h* and the biomass, lipids, and lipid soluble parameters were evaluated in each medium. All these results are introduced in Appendix A. The values are means ± SD (n = 3).

The production of biomass was generally higher in R. glutinis CCY 20-2-26 and S. pararoseus CCY 19-9-6 than in other strains. The highest production of biomass was found in media with hydrolyzed fat in all strains except C. macerans CCY 10-1-2. Emulsified fat was a better substrate than crude fat.

The production of carotenoids on glucose regularly grew with the cultivation time in all strains; the yields of total carotenoids were in range of 2.0–5.8 mg/L and, as the best producers, *R. glutinis* CCY 20-2-26 and *S. pararoseus* CCY 19-9-6 were evaluated (Appendix A). More fluctuations in carotene yields were observed during stationary growth on glycerol in all strains, which corresponds with supposed osmotic stress role of glycerol in red yeasts [18,19]. More stable yields were obtained in crude fat medium. The final yields of carotenoids were similar in both *Rhodotorula* strains on glucose, glycerol and crude fat. *R. glutinis* CCY 20-2-26 was a substantially better producer of carotenes than *R. mucilaginosa* CCY 19-4-6. A high increase of carotenoid formation on crude fat medium was observed for *C. macerans* CCY 10-1-2 and *S. pararoseus* CCY 19-9-6; the yield was about 2–3× higher than in the glucose medium. *S. pararoseus* CCY 19-9-6 was the best producer of carotenoids on fat substrate, with the yield of 13.4 mg/L of culture. The production of carotenes was the highest in *R. glutinis* CCY 20-2-26 and an induction of pigment production was observed in all fat media (mostly in the hydrolyzed fat medium).

The production of ergosterol was quite stable during the stationary phase and similar in all strains. A slight increase of ergosterol concentration with the cultivation time could be observed in most strains and the values ranged from 0.2 to 0.96 mmol/L. Stable production is probably accompanied by the need for stable membrane structures in stationary cells.

Similarly to ergosterol, the production of ubiquinone was relatively stable. The highest production of ubiquinone, about 3–5× higher than in other strains, was found in S. pararoseus CCY 19-9-6. In this strain, as well as in the others, several times higher production of ubiquinone on fat media, when compared with glucose, was observed. The reason is probably in the changes of the oxidation conditions; more effective oxygen utilization and higher biomass production is connected with intensive respiration by cells.

Lipid production during growth increased gradually with the time of cultivation in all strains. All strains produced a 1.3–2× higher amount of total lipids (Appendix A) in the glucose medium compared with glycerol. Rhodotorula strains produced a similar amount of total lipids on all three media. In R.mucilaginosa CCY 19-4-6 and S.pararoseus CCY 19-9-6, quite higher total lipids were produced on fat than on glucose, which were significantly higher than on glycerol medium. Production of larger amount of lipids in these two strains is probably interconnected with the induction of extracellular lipase and biosurfactant production, which, alltogether, could represent a complex metabolic response to the fat medium. 

We can summarize that under C/N = 13, red yeast cells growing on fat-based media exhibited higher biomass production and accumulated some metabolites, especially in media containing hydrolyzed fat and emulsified fat.

#### 3.5.2. Production of Metabolites on Fat-Based Media at Different C/N Ratios

In following part of this work we studied the effect of the C/N ratio on the biomass yield and metabolite production for yeasts grown for 96 h (see Section 2.3.2, c)). The results of biomass and metabolite yields on different fat-based and control media, and for four C/N ratios from 13 to 100, are presented in Figure 6 (values are means ± SD (n = 3)). Some results regarding the influence of the C/N ratio on the metabolic activity are also shown in Figure 5 (see Section 3.4), where increasing the C/N ratio from 13 to 50 caused a several times decrease of biosurfactant production. 

Changes of growth and biomass production differed in individual strains according to the medium composition and C/N ratio. While growth on glucose w growing C/N ratio was quite stable in R. glutinis CCY 20-2-26, R. mucilaginosa CCY 19-4-6, and S.pararoseus CCY 19-9-6, in C. macerans CCY 10-1-2 increasing the C/N ratio to 50 led to some increase in the biomass yield. Compared with glucose, in some strains, the growth on glycerol as a substrate led (R. glutinis CCY 20-2-26, S. pararoseus CCY 19-9-6) to lower biomass production at C/N 13 and a gradual decrease to C/N 50. Oppositely, in C.macerans CCY 10-1-2 and R. mucilaginosa CCY 19-4-6, an increased biomass was produced with a higher C/N (Figure 6a). A several times increase in biomass production was observed on the crude fat (namely in R. mucilaginosa and C.macerans). In fat-based media, a higher C/N ratio was needed for a higher biomass production, especially in the crude fat medium (Figure 6a).

As expected, in all strains, production of lipase (Figure 6b) on glucose was similar and independent of the C/N ratio. In the glycerol medium, lipase production was induced by C/N to 50 in both Rhodotorula strains, probably due to the presence of glycerol as the lipid degradation product. Extracellular lipase was induced in all strains cultivated on crude fat media, especially at lower C/N ratios (13–50). Emulsified fat led to a generally lower production of lipase. Induction was observed in C. macerans CCY 10-1-2 and mainly in S.pararoseus CCY 19-9-6 up to C/N 100.

In all strains, production of the percentage of total lipids per CDW increased in glucose media with a growing C/N ratio to C/N 50; a further increase in C/N had a slightly negative or no effect on lipid formation (Figure 6c). In glycerol medium, a similar trend as on glucose was observed. Nevertheless, the amount of lipids produced was lower. In fat media, the yield of lipids increased slightly to C/N 25 in C. macerans CCY 10-1-2 and R. mucilaginosa CCY 19-4-6 and to C/N 50 in R. glutinis CCY 20-2-26, while in S. pararoseus CCY 19-9-6 about a 4× increase was found in C/N 25, then lipid formation gradually decreased. The strain S. pararoseus CCY 19-9-6 was the best producer of lipids on crude fat and a C/N ratio 25–50 had the highest yield (53.2% of lipids/CDW). Similarly, a C/N ratio of 25 was optimal for the production of lipids on fat in C. macerans CCY 10-1-2 (47% of lipids/CDW) and R.glutinis CCY 20-2-26 (26% of lipids/CDW).

Production of carotenoids (Figure 6d) dramatically decreased (3–10×) with an increased C/N ratio in all strains, with exception of R. mucilaginosa CCY 19-4-6 at C/N 50. R. glutinis CCY 20-2-26 and S. pararoseus CCY 19-9-6 showed gradually lower carotenoids values with increased C/N ratios in glycerol media (3–4×). In other strains, slightly changed values with a slight increase at C/N 50 were recorded, followed by a decrease of carotenoid content at C/N 100. In the crude fat medium, a repeatly dramatic decrease of carotenoid content with an increased C/N ratio was observed in R. glutinis CCY 20-2-26 and in S. pararoseus CCY 19-9-6. In C. macerans CCY 10-1-2, a slight increase of carotenoid formation at C/N 50 was found, while in R. mucilaginosa CCY 19-4-6 a gradual increase of carotenoids with an increased C/N ratio was observed. In R. glutinis CCY 20-2-26 and S. pararoseus CCY 19-9-6 cultivated on emulsified fat, a high carotenoid content was measured at C/N 13, then there was a strong decrease at C/N 25, and a further increase was found at higher C/N ratios. In the other two strains, a strong increase at C/N 25 and then a gradual decrease at higher C/N ratios was observed.

Production of ubiquinone in all strains decreased with a growing C/N ratio in glucose medium (Figure 6f). The production of ubiquinone on the glycerol medium exhibited similar tendencies in S. pararoseus CCY 19-9-6 and R. mucilaginosa CCY 19-4-6, while in R. glutinis CCY 20-2-26 and C. macerans CCY 10-1-2 some fluctuation are observed at higher C/N values. On crude fat, a decrease of ubiquinone with a growing C/N was found, especially in S. pararoseus CCY 19-9-6 and R. glutinis CCY 20-2-26. In the other two strains, an increase of ubiquinone production was observed. In C. macerans CCY 10-1-2 about a 15× increase was found at C/N 25 and and a 9× increase was foind at C/N 50, respectively. The medium with emulsified fat led to similar, but more moderate, effects as in the previus case (see Figure 6).

The production of ergosterol exhibited similar trends in all strains (Figure 6e). A decrease of ergosterol formation was accompanied with increased C/N ratio mainly in R. glutinis CCY 20-2-26 and S. pararoseus CCY 19-9-6, and also at higher C/N ratio in R.mucilaginosa CCY 19-4-6.

#### 3.5.3. Fatty Acid Composition in Fat-Based Media

During measurement of the total lipids, the composition of fatty acids depending on the medium composition was also followed. Figure 7 shows the fatty acid composition of the yeast strains cultivated for 96 h in processed fat media (according to Section 2.3.2, b). In this figure, the ratio of SFA:MUFA:PUFA is illustrated. According to analysis of individual fatty acids, in all strains the major representatives of saturated fatty acids (SFA) were palmitic acid (C16:0) and stearic acid (C18:0). Monounsaturated (MUFA) oleic acid (C18:1n9c) and polyunsaturated fatty acids (PUFA) were represented mainly by derivatives of linoleic acid (C18:2n6c). In all the species, the most abundant fatty acid present was oleic acid, with a production of over 35–45%. The other fatty acids, namely linolenic acid (C18:3n3c), were present in small amounts. Data are evaluated as the ratio of the sum of SFA, MFA (oleic acid), and the sum of PUFA and other rarely present acids. According to data obtained by GC/FID (see Section 2.4.5), changes in fatty acid composition were observed depending on the yeast strain and the composition of the cultivation medium.

In the glucose medium, the highest amount of oleic acid (63–81%) was found in all strains. Changes of C-source to glycerol caused different distributions of individual fatty acids in microbial oil. The portion of PUFA increased 3–4× in all strains and oleic acid was proportionally decreased. In fat media, more strain specific differences of fatty acid (FA) composition could be found. A higher degree of PUFA in the crude fat medium was found mainly in C. macerans CCY 10-1-2 (37%) and R. glutinis CCY 20-2-26 (33%). The emulsifiation of fat led to higher degree of PUFA in C. macerans CCY 10-1-2, up to 44%, while the rest of the strains contained abround 20% PUFA. Composition of FA in yeast cells cultivated on fat hydrolyzate was similar to on the glycerol medium; the content of PUFA ranged from 14% to 25%. Thus, C. macerans CCY 10-1-2 cultivated on fat medium can be used for very effective valorization of animal waste fat to microbial oil, with a high content of essential polyunsaturated fatty acids.

#### 3.5.4. Production of Glucans

Polysaccharides belong to the group of stress-connected metabolites and are overproduced at different stress conditions [24]. They are formed and accumulated extracellularly, as well as inside or on the surface of the cells [25]. Cultivation of tested red yeasts in the control glucose production medium for 96 h led to a relatively low accumulation of total glucans in all the tested strains (*S. pararoseus* CCY 19-9-6, *R. glutinis* CCY 20-2-26, *R. mucilaginosa* CCY 20-9-7, and *C. macerans* CCY 10-1-2.). Due to the fact that the C/N ratio is a well-known parameter that influences cellular metabolism and accumulation of different metabolites, as well as the biomass yield [25] in yeasts, we studied the effect of the C/N ratio on the total glucan production for 96 h in comparison with the total lipid production in the same conditions.

The resulting glucans and the sum of the glucan and lipid yields are presented in Table 5. The significant increase in glucans content under high C/N conditions was not observed in the tested strains. Oppositely, all strains exhibited a slight decrease of total glucan content with increased C/N ratios. It is well known that a high C/N ratio can have a positive effect on the accumulation of some intracellular metabolites (glucans, lipids, pigments, etc. [8,25,26,27]). Thus, the increase in the biomass yield under a high C/N ratio (see Section 3.5.1) could be a result of mainly higher accumulation of certain metabolites; in our case glucans and lipids, but not because of the increase in cell proliferation. In Table 5 we illustrate this fact by the introduction of the summary amount of glucans and lipids accumulated in the biomass at the growing C/N ratio. With exception of the lowest C/N value, when the ratios of glucans and lipids were quite differed in individual strains, at the higher C/N we observed quite stable values of glucans as well as the typical sums of glucans and lipids for individual strains. The sum of glucans and lipids formed 50–60% of the biomass and changed only slightly with a further increase of the C/N ratio.

The simultaneous production of glucans and lipids as stress-connected metabolites in relation to nutrition stress and increased C/N ratios is also documented in Appendix A and Figure 8. Appendix A presents electron microscopy images of red yeasts cultivated on glucose at lower (C/N = 13) and higher (C/N = 100) C/N ratios. On the surface of cells, some newly formed structures can be observed. We can suppose there are some types of polysaccharides (e.g., glucans) which are bound to the cell wall and can be detected as cellular components [24,25]. Changes in cell wall composition belong to the general mechanisms of cell response in yeasts. In Figure 8 we present transmission electron microscopy images of red yeast cells cultivated at the same conditions as above. Lipid bodies can be observed inside the cells cultivated at higher C/N ratios.

For illustration, the complete characteristics of the red yeast biomass produced in fat containing media at different C/N ratios are summarized in Appendix A.

## 4. Discussion

Oleaginous fungi are exceptional organisms for producing high-value multifunctional biomasses from agriculture supply chain by-products, since they can utilize a wide range of growth substrates, including wastes. Oleaginous fungi, both yeast and filamentous fungi, are able to produce and accumulate up to 85% (*w*/*w*) of lipids [26]. The produced lipids mainly consist of triacylglycerols (TAGs). Due to their versatile metabolic system, different oleaginous fungi are able to co-produce a wide range of chemicals in addition to lipids, as follows: (i) Oleaginous red yeasts are capable of performing simultaneous co-production of β-glucans [24], carotenoid pigments [26], proteins, and lipids [27]; and (ii) oleaginous filamentous fungi are capable of performing simultaneous co-production of lipids, pigments, and bio-polymers [23,28]. Thus, oleaginous fungal biomass could be directly used as a high-value multifunctional biomass in feed and food nutrition, pharmacology, and medicine. Alternatively, the biomass could be separated and utilized in different markets that seek certain specific functionalities [29,30].

A decreasing availability of one substrate to oleaginous yeasts can, in many instances, be compensated by the utilization of another [31,32,33,34,35]. Additionally, in our previous studies, red yeast strains were screened for waste glycerol and animal fat utilization [18,19]. To the best of our knowledge, this study focused on the uptake of solid animal fat was the first report about the ability of red yeasts to metabolize crude fat [18]. Extreme differences in biomass production on glucose and fat were observed using flow cytometry experiments. This technique is probably not suitable for biomass determination using non-usual substrates. As another screening technique, the drop test indicated that red yeasts are able to utilize crude animal fat and products in its chemical hydrolysis. In this test different utilizations of glycerol vs. glucose can be observed as typical examples of strain specific stress response to glycerol concentration [19]. Preliminary cultivations of some strains in liquid medium with combined C-sources showed that the tested strains grew sufficiently on a fat with the addition of glucose, while cultivation without glucose led to very low biomass formation [18]. The most promising strains were then selected for the present metabolic study.

In this study we found and verified that all the tested red yeast strains were able to utilize animal waste fat (solid/crude and processed), as well as glycerol, and produce biomass to a large extent. Biomass production was strain-specific, depending on type of medium. While growth on glucose and glycerol was relatively similar in all strains, fat media caused larger differences between individual strains. Similar to in the screening study, the amount of biomass produced on glycerol was about 20% lower than on glucose. Glycerol is probably the main carbon source for yeast cells after full or partial fat hydrolysis. All fat-containing media were utilized by red yeasts to a similar or slightly higher degree than glycerol alone, with the exception of *S. pararoseus* CCY 19-9-6. All strains exhibited the best growth on hydrolyzed fat. As an additional C-source, fatty acids formed during fat hydrolysis can be considered.

Glycerol was mentioned above as the most often utilized carbon source in processed fat media [19]. In recent study [35], it was found that red yeasts growing on glycerol preferably synthesized less amino acids, which resulted in a higher availability of acetyl-CoA to enter the carotenoid and fatty acid synthesis pathways. This could contribute to the higher levels of carotenoids and total intracellular fatty acids observed in some strains grown on glycerol, and probably the fat-based media as well. In some studies, the pH measurements of the growth media of some fungi have shown a rapid drop of pH values as a result of the simultaneous production of acids and lipids, although a general down-regulation of the TCA cycle was mostly observed [30]. It has been previously reported that some oleaginous fungi can produce large amounts of organic acids (mainly citric acid), at the expenses of lipid accumulation, when grown under nitrogen-limited conditions [17,22,30].

Thus, in the present study, the pH of the medium was monitored in all cultivations. While growth on glucose led to a significant decrease of pH in all strains, pH was slightly fluctuated or unchanged during the whole growth in glycerol and fat-based media. The decrease of pH is probably interconnected with carbon flux to citric acid production, which is the highest in glucose medium [30,31]. Adaptation to glycerol and fat media is probably not directly accompanied by citric acid production and, thus, other types of metabolic adaptation can be expected. Another explanation is an enhancement of the citric acid concentration, which can be achieved in the higher aeration condition. As a consequence of the presence of solid components in the medium, less aeration can be suspected in flasks with fat media and, thus, pH decrease is not so dramatic as in a glucose medium [17].

In lipid-containing media, induction of extracellular lipase can be expected as an adaptation mechanism for fat substrate utilization [29]. Lipase production largely depends on microorganisms, culture practices, medium composition, and cultivation design. Among the industrial waste residues used for lipase production, by-products of oil extraction from seeds and oil refining processes, as well as solid lipid wastes, can be used [28,29]. Our study confirms the production of extracellular lipase is induced by the presence of animal fat in the cultivation medium.

In previously published studies, the simultaneous production of citric acid and lipase was also observed [17,29]. Some authors describe the maximum lipase production to be in the exponential phase of growth, while citric acid synthesis was enhanced only in the stationary phase [7,9,11]. In our study, the phase of lipase induction seemed to be strain/medium specific. The reason is probably strain specific utilization of endogenous substrates during exponential and prolonged stationary phases [19].

A biorefinery concept can be proposed for the complex utilization and valorization of crude solid animal fat as a sole C-source. Another way is the hydrolysis of waste fat to glycerol and fatty acids or fatty acid salts. In our previous screening study [18], two types of chemical hydrolysis were tested for animal fat hydrolysis and media preparation. Saponification led to the formation of glycerol and fatty acids salts, exhibited high yields of hydrolysis, and a quick extraction and separation of fatty acids, but was less suitable for microorganism cultivation, due to the large salt concentrations in the cultivation medium. The products of acid hydrolysis are glycerol and free fatty acids and the hydrolysate should be neutralized before further use [18]. In this study, for the first time, enzyme hydrolysis by a microbial lipase was used for fat processing in red yeast media.

The addition of an emulsifier could also increase the bioavailability of the solid waste fat to the yeast cells, as demonstrated in *Yarrowia lipolytica* [9,10,11]. Biological emulsifiers, biosurfactants, are often produced simultaneously with lipases by the microorganisms generally involved in the metabolization of oil substrates [24,36]. However, the relationship between the production of lipases and biosurfactants has not been established yet. Biosurfactants are widely known surface active agents which comprise a wide range of chemical structures. Typical for *Basidiomycetes* (red yeasts), esters of polyols and fatty acids (PFAE), containing no sugar components, were found [36]. In this study, simultaneous production of lipase and biosurfactants was predominantly observed in *C. macerans* CCY 10-1-2 and *S. pararoseus* CCY 19-9-6 strains on the fat substrate. While these strains belong to the best producers of lipids on crude fat substrates, this is probably a mechanism of adaptation of these yeasts to lipid media and a transformation of exogenous waste lipids to microbial lipids with different fatty acid compositions. Of interest is the influence of the C/N ratio on biosurfactant production by red yeasts, as with a growing C/N ratio a dramatic decrease of biosurfactant production was observed; the explanation for this should be found in further studies.

During cultivation to the stationary phase on C/N 13, red yeast cells growing on fat-based media exhibited higher biomass production and metabolites accumulation, especially in media containing hydrolyzed fat and emulsified fat than in glucose/glycerol media. The production of lipids and lipid-soluble metabolites was enhanced by several times; the highest yields of carotenoids (24.8 mg/L) and lipids (54.5% of CDW) were found in the S. pararoseus CCY 19-9-6 strain. Simultaneous production of lipase and biosurfactants was observed in C. macerans CCY 10-1-2 and S. pararoseus CCY 19-9-6 on fat substrate; these strains belong to the best producers of lipids. Metabolites produced to a higher degree were lipids, carotenoids, and ubiquinone and these could be involved in the complex response of yeast cells to nutrition stress and fat substrate as well. The reason for the several times higher production of ubiquinone in all strains on fat media compared with glucose is probably because of changes of oxidation conditions, more efficient oxygen utilization, and higher biomass production connected with intensive cell respiration. As a consequence of the higher production of ROS (Reactive Oxygen Species) under these conditions, an extremely high production of carotenoids as membrane antioxidants occurred in some strains during growth in the crude fat medium. Thus, the coordinated high production of lipid-soluble metabolites can serve as an cross-response adaptation mechanism to nutrition stress. Strains R.muciulaginosa CCY 19-4-6 and S.pararoseus CCY 19-9-6 produced substantially higher amounts of lipids in fat-based media, which is probably interconnected with the induction of extracellular lipase and biosurfactant production. These processes alltogether could represent the complex metabolic response to the fat medium.

Due to the fact, that the C/N ratio is a well-known parameter influencing yeast cellular metabolism and the production of different metabolites, as well as the biomass yield [27], we studied the effect of four C/N ratios from 13 to 100 in different fat-based and control media on the growth and metabolic activity of tested red yeasts. From growth data, we concluded that in fat-based media a higher C/N ratio is needed for higher biomass production, especially in the crude fat medium. Extracellular lipase was induced in all strains cultivated on crude fat media, especially at lower C/N ratios (13–50). Emulsified fat led to generally lower production of lipase. Induction was observed in C. macerans CCY 10-1-2 and mainly in S.pararoseus CCY 19-9-6 up to C/N 100. Additionally, in glycerol medium, lipase production was induced by C/N to 50 in both Rhodotorula strains, probably due to the presence of glycerol as the lipid degradation product. The production of total lipids increased several times in all strains to C/N 50. Oppositely, the production of carotenoids, ubiquinone, and ergosterol dramatically decreased for several times with an increased C/N ratio in all strains. Based on the complex data obtained, it can be concluded that carotenoids are produced oppositely to lipids, dependent on the C/N ratio. This could be caused by competition of availability of acetyl-CoA as a substrate both for lipids (fatty acids) and isoprenoids (carotenoids).

In all strains, the production of ubiquinone exhibited similar trends as the production of carotenoids; the simultanous decrease of both parameters with a growing C/N was found, especially in R. glutinis CCY 20-2-26 and S. pararoseus CCY 19-9-6. The reason is probably the production of both metabolites by the isoprenoid pathway, and moreover, probably also concerns the involvement of both metabolites in the regulation of the redox state of the cell. Production of ergosterol also exhibited similar trends in all strains. This fact can be considered as evidence of the previously described simultaneous regulation of sterol and carotenoid branches in the isoprenoid pathway in red yeasts [18,19]. A decrease of ergosterol formation is accompanied by an increased C/N ratio, mainly in R. glutinis CCY 20-2-26 and S. pararoseus CCY 19-9-6, and also in R.mucilaginosa CCY 19-4-6 at higher C/N ratios. Decreased production of ergosterol may induce plasmatic membrane remodeling and changes in their permeability, which could lead to increased transport of substrates into the cells and, thus, increased accumulation of lipids in the cell mass.

Furthermore, fatty acid composition was measured in yeast cells according to the C-source in the medium. Changes of FA distribution probably belong to the complex response to nutrition stress and were described also by other authors [9,10,11,12,22,35]. An increased desaturation degree of fatty acids, commonly with decreased ergosterol production, can contribute to membrane permability changes. Oppositely, lower values of antioxidants can be caused by their increased uptake in oxidation stress or during cross-protection to other types of stress present in the cells during adaptation to unusual substrates [19]. Thus, C. macerans CCY 10-1-2 cultivated on a fat medium can be practically used for imensely effective valorization of animal waste fat to microbial oil with a high content of essential polyunsaturated fatty acids.

From the response obtained for all the experiments it is clear that microbial lipids and lipid-soluble metabolite production was dependent on the combination of the various factors studied. The microbial lipid content accumulated by red yeast cells ranged from 8% to 53.4% (mass of lipids per cell dry weight), demonstrating the potential of some red yeast strains for industrial production of microbial oils from animal waste fat (Appendix A). Accumulation of lipids on glucose media is mainly related to the C/N ratio and intracellular carbon flux to fatty acid synthesis, while in fat-based media changes in ergosterol production and fatty acid composition may influence membrane permeability and better synthesis of lipids from diacylglycerols [11]. The growth and maximum lipid production obtained in red yeasts are comparable with others found in the literature for lipid synthesis by *Y. lipolytica* strains from stearin [7] or commercially refined olive oil [10] and pork lard [17], and higher than that obtained with pure olive oil [9]. The only directly comparable study to the present work is the cultivation of *Y. lipolytica* on pork lard [17]. Depending on the cultivation conditions, the production of biomass in this study ranged from 4–8 g/L and lipid production reached 26–58% [17]. *Y. lipolytica* has been used as a model microorganism for the ex novo lipid accumulation, since it can “upgrade” the composition of fatty materials utilized as substrates with concomitant production of tailor-made lipids of high added value. Based on our findings, similar processes could probably also occur in oleaginous red yeasts.

There is very limited information regarding glucan content in red carotenogenic yeasts. When a growth-required nutrient is limited in the culture media, and the carbon is still available, the biosynthetic pathways of carbon-type metabolites (lipids, polysaccharides) in microorganisms are triggered [8,25,29,35]. Nevertheless, the significant increase in glucan content under high C/N conditions was not observed in the tested strains. Oppositely, all strains exhibited a slight decrease of total glucans content with an increased C/N ratio. At the higher C/N we observed quite stable values of glucans, as well as the sum of glucans and lipids typical for individual strains. The percentage of minor lipid-soluble compounds in red yeast biomass is relatively low and changes are strain-specific depending on the type of C-source. Nevertheless, red yeasts can be used as single cell biotechnological factories for the production of enriched high-value-added biomass (Appendix A).

This study would like to demonstrate that the cellular response to solid fat as a substrate is very complex and many different mechanisms are involved within. We tried to open some more windows, discover new options, and open new topics for cells stressed by nutrient limitation. Further experiments remain to be performed in order to better understand the overall adaptation mechanisms of red yeast cells.

## 5. Conclusions

In this work, four selected red yeast strains (*Cystofilobasidium macerans* CCY 10-1-2, *Rhodotorula glutinis* CCY 20-2-26, *Rhodotorula mucilaginosa* CCY 19-4-6, and *Sporobolomyces pararoseus* CCY 19-9-6) were cultivated in media containing crude, emulsified, and enzymatically hydrolyzed animal waste fat, compared with glucose and glycerol as single C-sources. All red yeast strains are able to utilize solid and processed fat. Cell growth and biomass production at C/N = 13 was higher in emulsified and hydrolyzed fat media than on glucose/glycerol. The production of lipids and lipid-soluble metabolites in fat-based media was enhanced by several times; the highest yields of carotenoids (24.8 mg/L) and lipids (54.5% of CDW) were found in the *S. pararoseus* CCY 19-9-6 strain. Simultaneous production of lipase and biosurfactants was observed in the strains that were evaluated as the best producers of lipids. An increased C/N ratio (13–100) in fat media led to higher biomass and total lipid production. Extracellular lipase was induced in all strains cultivated on crude fat media at lower C/N. Oppositely, production of carotenoids, ubiquinone, and ergosterol dramatically decreased by several times with an increased C/N ratio in all strains.

We can conclude that red yeast cells are able to adapt on solid fat media and to nutrition stress/limitation by several mechanisms, which are regulated simultaneously. Generally, fat-based media can be utilized well, especially in emulsified or hydrolyzed form. With the exception of cultivation conditions such as oxygen demand, temperature, and permanent lighting, the production of specific stress metabolites can help yeast cells survive and metabolize on non-usual waste substrates. Based on this study, sterols, carotenoids, ubiquinone, glycerol, polysaccharides, lipids, and fatty acids can be considered as stress metabolites and mechanisms involved in complex cross-protection of red yeast cells to diverse types of stress factors. Furthermore, in yeast cells grown on fat-based media, the induction of lipase and biosurfactants was observed. Most of the above-mentioned metabolites can be used in many important industrial applications, mainly in the food and feed industry and in pharmacology.

## Figures and Tables

**Figure 1 microorganisms-07-00578-f001:**
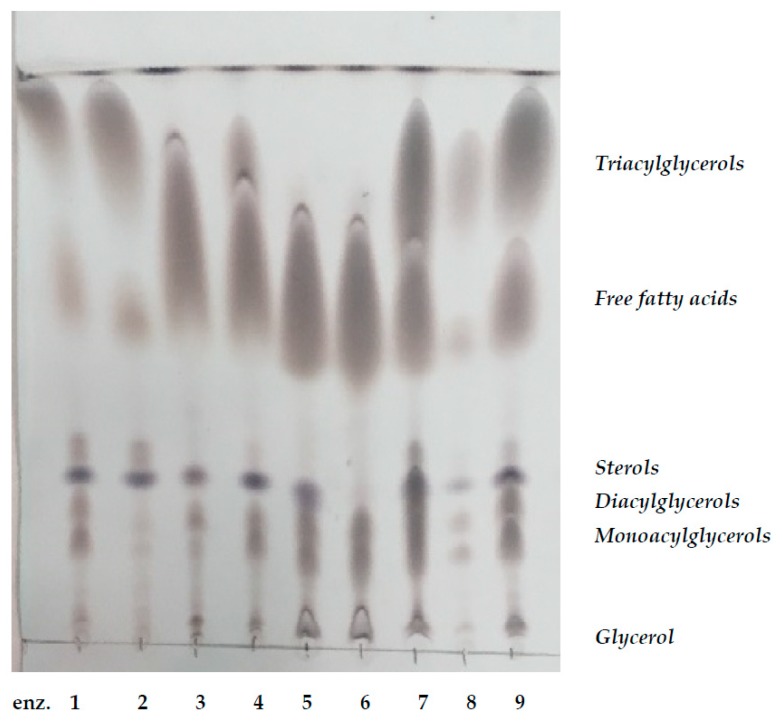
TLC chromatography of samples of animal fat hydrolyzed by different lipases. Enz., enzyme. Numbers—the numbers of the individual enzymes are introduced in Table 4.

**Figure 2 microorganisms-07-00578-f002:**
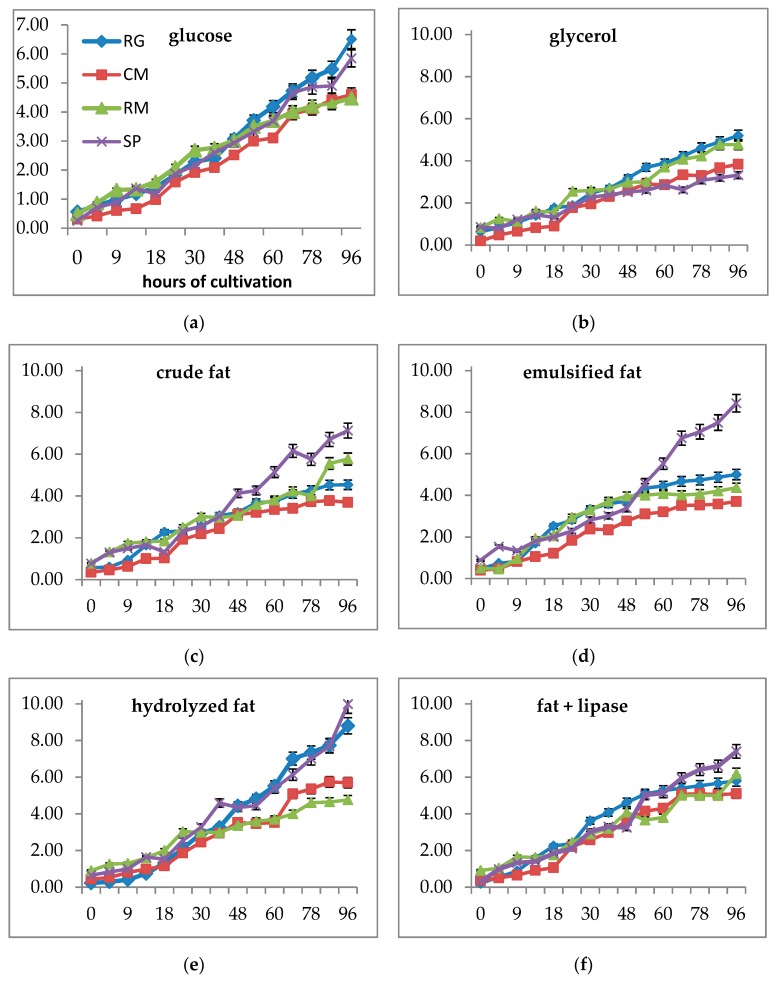
Growth curves of yeast strains cultivated in different media for 96 h. Biomass production in g/L was measured in the following media: (**a**) Medium with glucose; (**b**) medium with glycerol; (**c**) medium with crude animal fat; (**d**) medium with emulsified fat by Tween 80; (**e**) medium with enzymatically hydrolyzed fat; and (**f**) medium with animal fat and the addition of lipase. Abbreviations: CCY (Culture Collection of Yeasts), RG (*Rhodotorula glutinis* CCY 20-2-26), CM (*Cystofilobasidium macerans* CCY 10-1-2), RM (*Rhodotorula mucilaginosa* CCY 19-4-6), and SP (*Sporobolomyces pararoseus* CCY 19-9-6).

**Figure 3 microorganisms-07-00578-f003:**
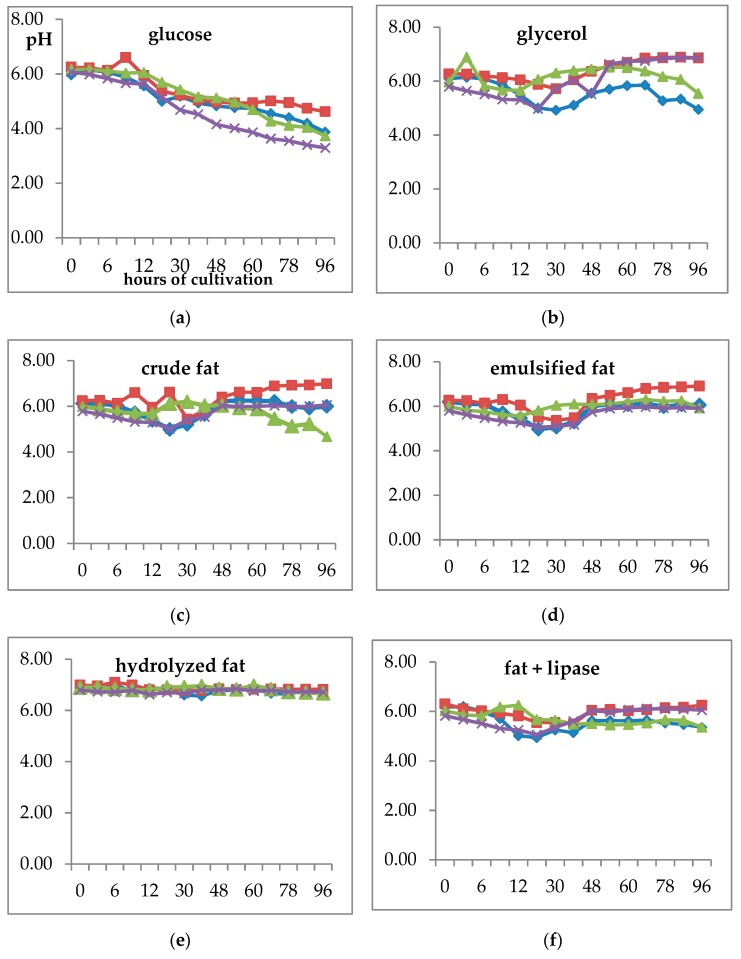
Changes of pH of individual media during yeast cultivation. Changes of pH were recorded in following media: (**a**) Medium with glucose; (**b**) Medium with glycerol; (**c**) Medium with crude animal fat; (**d**) Medium with emulsified fat by Tween 80; (**e**) Medium with enzymatically hydrolyzed fat; (**f**) Medium with animal fat an addition of lipase. Abbreviations: RG (*Rhodotorula glutinis* CCY 20-2-26), CM (*Cystofilobasidium macerans* CCY 10-1-2), RM (*Rhodotorula mucilaginosa* CCY 19-4-6) and SP (*Sporobolomyces pararoseus* CCY 19-9-6).

**Figure 4 microorganisms-07-00578-f004:**
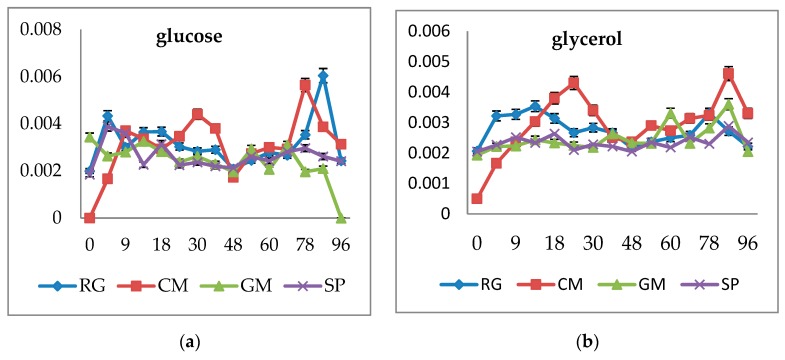
Changes of lipase production of individual strains during cultivation on different media. Lipase production in U/mL of medium was measured in the following media: (**a**) Medium with glucose; (**b**) medium with glycerol; (**c**) medium with crude animal fat; (**d**) medium with enzymatically hydrolyzed fat; (**e**) medium with animal fat and the addition of lipase. The production of lipase on the emulsified medium was not detected. Abbreviations: RG (*Rhodotorula glutinis* CCY 20-2-26), CM (*Cystofilobasidium macerans* CCY 10-1-2), RM (*Rhodotorula mucilaginosa* CCY 19-4-6), and SP (*Sporobolomyces pararoseus* CCY 19-9-6).

**Figure 5 microorganisms-07-00578-f005:**
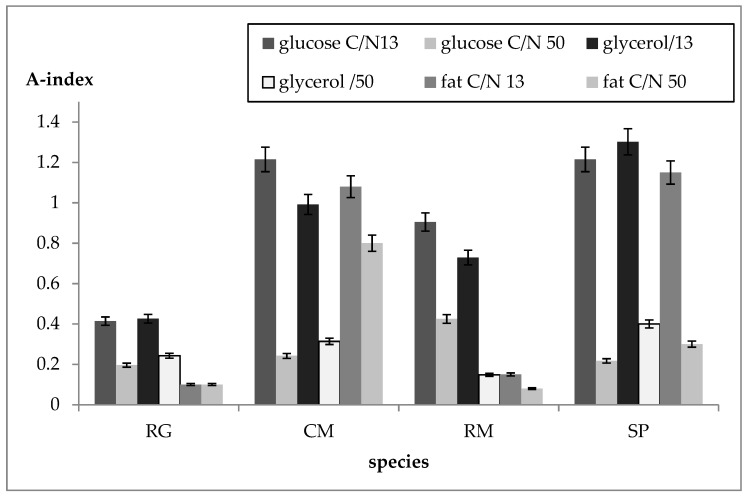
Production of biosurfactants by individual strains at different C/N ratios. Abbreviations: RG (*Rhodotorula glutinis* CCY 20-2-26), CM (*Cystofilobasidium macerans* CCY 10-1-2), RM (*Rhodotorula mucilaginosa* CCY 19-4-6) and SP (*Sporobolomyces pararoseus* CCY 19-9-6); A-index was evaluated as absorbance at 345 nm when compared with blank.

**Figure 6 microorganisms-07-00578-f006:**
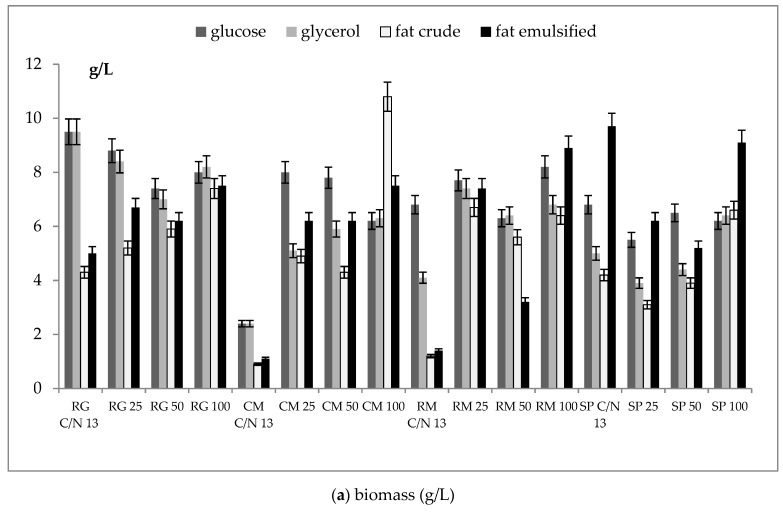
Production of biomass and metabolites at different C/N ratios. Changes of biomass production (**a**) in RG (*Rhodotorula glutinis*), CM (*Cystofilobasidium macerans*), RM (*Rhodotorula mucilaginosa*), and SP (*Sporobolomyces pararoseus*) in different media—glucose, glycerol, crude fat, emulsified fat; (**b**) Lipase production; (**c**) Lipids production; (**d**) Production of carotenoids; (**e**) production of ergosterol; (**f**) Production of ubiquinone. Abbreviations: RG (*Rhodotorula glutinis* CCY 20-2-26), CM (*Cystofilobasidium macerans* CCY 10-1-2), RM (*Rhodotorula mucilaginosa* CCY 19-4-6), and SP (*Sporobolomyces pararoseus* CCY 19-9-6).

**Figure 7 microorganisms-07-00578-f007:**
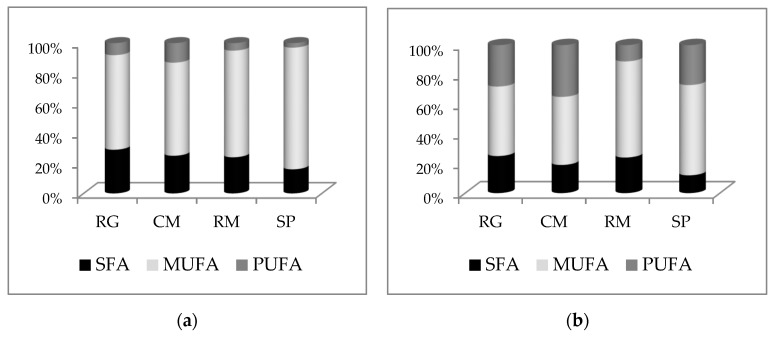
Fatty acid composition of total lipid fraction after 96-h cultivation in different media. (**a**) Medium with glucose; (**b**) Medium with glycerol; (**c**) Medium with crude animal fat; (**d**) Medium with emulsified fat; (**e**) Medium with enzymatically hydrolyzed fat. Abbreviations: RG (*Rhodotorula glutinis* CCY 20-2-26), CM (*Cystofilobasidium macerans* CCY 10-1-2), RM (*Rhodotorula mucilaginosa* CCY 19-4-6) and SP (*Sporobolomyces pararoseus* CCY 19-9-6). SFA, saturated fatty acids; MUFA, monounsaturated fatty acids; PUFA, polyunsaturated fatty acids.

**Figure 8 microorganisms-07-00578-f008:**
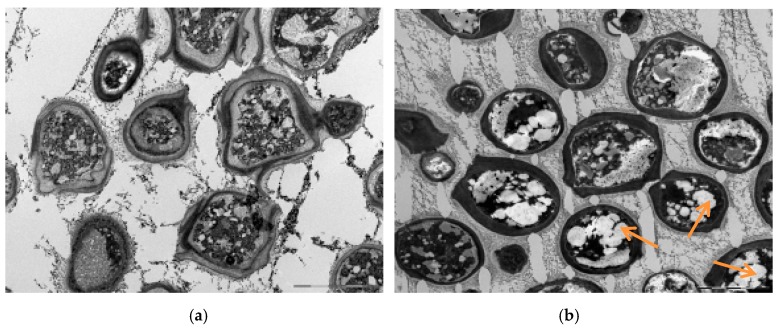
Transmission electron microscopy of red yeast cells cultivated on glucose at different C/N ratios. Formation of lipid bodies under nutrition stress. Lipid bodies are demonstrated by arrows. (**a**) *R. glutinis* CCY 20-2-26 C/N 13; (**b**) *R. glutinis* CCY 20-2-26 C/N 100; (**c**) *C. macerans* CCY 10-1-2 C/N 13; (**d**) *C. macerans* CCY 10-1-2 C/N 100; (**e**) *S. pararoseus* CCY 19-9-6 C/N 13; and (**f**) *S. pararoseus* CCY 19-9-6 C/N 100.

**Table 1 microorganisms-07-00578-t001:** Composition of production media with different C/N ratios.

C/N Ratio	Medium Component			
	Glucose	Glycerol	Waste Fat	Waste Fat/Tween 80
**13**	30.00	30.67	18.7	18.7/1.87
**25**	49.50	46.00	35.00	35.00/3.50
**50**	99.90	93.00	70.00	70.00/7.00
**100**	199:80	186.00	140.00	140.00/14.00

**Table 2 microorganisms-07-00578-t002:** Elution program for high-performance liquid chromatography (HPLC).

Time (min)	Mobile Phase A (%)	Mobile Phase B (%)
0.0	100.0	0.0
10.0	0.0	100.0
14.0	0.0	100.0
15.3	100.0	0.0
19.0	100.0	0.0

**Table 3 microorganisms-07-00578-t003:** Temperature program of GC analysis.

	Retention Time (min)	Gradient (°C·min^−1^)	Temperature (°C)	Retention (min)
**1**	0.000	start	-	-
**2**	1.000	0.000	80.000	1.000
**3**	5.000	15.000	140.000	0.000
**4**	21.667	3.000	190.000	0.000
**5**	25.467	25.000	260.000	1.000
**6**	25.467	stop	-	-

**Table 4 microorganisms-07-00578-t004:** Time-course production of glycerol during hydrolysis of waste animal fat by lipases of different origin.

Lipase Source	Enzyme Nr	Activity U/mg of Protein	Glycerol (g/L) Per Time of Hydrolysis				
			1 h	3 h	6 h	24 h	30 h
***Aspergillus niger***	1	5–15	0.03	0.16	0.38	0.67	1.16
***B Candida antarctica*, recombinant from *Aspergillus oryzae***	2	9	0.46	1.18	1.28	1.61	2.28
***Candida rugosa***	3	2	1.06	9.75	13.52	14.9	15.85
***Mucor miehei***	4	1	0.64	1.52	7.23	11.05	11.84
***Pseudomonas cepacia***	5	30	0.21	0.78	4.48	5.63	6.69
***Pseudomonas fluorescens***	6	40	0.02	0.68	2.35	3.42	4.08
***Rhizopus oryzae***	7	30	0.20	0.78	1.34	1.42	1.82
***Rhizopus niveus***	8	1.5	0.02	0.24	0.49	0.51	0.64
***Porcine pancreas***	9	30–90	0.01	0.07	0.13	0.17	0.19

Glycerol concentrations are introduced as means (*n* = 2).

**Table 5 microorganisms-07-00578-t005:** Summary production of beta-glucans and lipids on glucose at different C/N ratios.

Strain	C/N 13		C/N 25		C/N 50		C/N 100	
	Glucans %	Sum G+L %	Glucans %	Sum G+L %	Glucans %	Sum G+L %	Glucans %	Sum G+L %
**RG**	23.18	32.35	23.80	56.65	21.05	58.37	20.35	56.87
**CM**	26.15	32.10	23.61	58.84	18.53	65.05	16.32	63.59
**RM**	17.52	30.24	14.60	46.79	15.54	49.77	15.99	49.25
**SP**	14.30	25.38	16.87	47.73	15.58	54.94	14.73	53.85

For this illustrative comparison, glucan concentrations are introduced as average values from 2 measurements as the percentage of CDW. Sum G+L is the summarized value of glucans (G) and total lipids (L) in the percentage of CDW. Average total lipid values were used (without SD) as introduced in Table 3. Abbreviations: RG (*Rhodotorula glutinis* CCY 20-2-26), CM (*Cystofilobasidium macerans* CCY 10-1-2), RM (*Rhodotorula mucilaginosa* CCY 19-4-6), and SP (*Sporobolomyces pararoseus* CCY 19-9-6).

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
