# Peer review of "Study of Metabolic Adaptation of Red Yeasts to Waste Animal Fat Substrate"

_microorganisms, 2019, doi:10.3390/microorganisms7110578_

Round 1

Reviewer 1 Report

The manuscript “Study of metabolic adaptation of red yeasts to waste animal fat substrate” is very rich in data and concerns a topic particularly interesting from a theoretical and applicative point of view. So congratulations!

But some critical aspects emerge during the reading.

English should be revised in all the text, especially in Results and Discussion paragraphs. There are many mistakes, typos, and some sentences are not correct and difficult to understand. (e.g. line 94 “rude” instead of crude; line 210 “at glucose medium” and “ To induction of biosurfactant production”, etc……) Results and, in particular, discussion are too long and prolix. The data are very interesting and meaningful but should be organized in a much more organic way. Short sentences with a specific final meaning are needed. In this form the reader gets lost into the large amount of data and results. E.g. lines from 650 to 709 should be shorted in 10 lines and so on…..

Other minor aspects to be considered:

Keywords: words present in the title can be deleted from keywords group Line 76: “Research and development of processes has so far been focused on the utilization of simple sugars…” This is not true, there are many papers concerning lipid production from complex carbon sources. Please correct this sentence adding some information at this point. Line 97 and 107. The authors used the word adaptation. Why do you use this concept? Did you see 2 different behaviors on yeasts growth? Lag and log phases? Lines 150 and 154. The authors talk about inocula and production medium but it is not clear if this media correspond to the medium describe at lone 157-158 and named inoculation medium. Clarify this point Lines 159-168. Which is the C/N values of all this media. This information should be added. Line 170. Why “ light exposure” was used? Clarify the reason Lines 178-180 + Scheme 1. It is not clear how the authors obtained the different C/N ratio media. What was the source of N? Why fat enzyme and hydrolyzed fat were not used? Paragraph 2.4. Considering that the different measurements in different media were obtained in different times, in order to show a clear plan of the experiments, the authors should mention when the single measurements have been done (a scheme can be useful) Line 237. “Lipids were extracted from 10 mg of freeze-dried cells”. How? The authors should explain how did they extract lipids. Paragraph 2.5. In which medium did the observed cells were grown? Lines 289-296. This part should be moved to Discussion Lines 297-309. This part was already explained in Materials and Methods. At least table 1 can be moved to this paragraph instead that left in Materials and Methods, and a short comment concerning the obtained results can be added. Lines 314-319. This part was already explained in Materials and Methods. Lines 328-331. Why did the authors stopped the test at 96h? 3 and 4 are clearer if the titles of the different tests are reported in figures (like done in Fig. 2) Line 383. “ induction of biomass”. What does it mean? Line 395. Why there’s no graph of this test (addition of emulsifier) Lines 398-400 and others points in the text. Why did the authors talk about stationary phase if they asserted that “ It is necessary to note that after 96 hours of growth the end of stationary phase probably was not reached in most of cultivations.” and it is clear by the graphs (fig. 2) that no stationary phase was reached? Line 431. “ (60-96 hours” )Why did the authors select this interval of time? Line 432. Tables 2-4 instead of 1-3 Table 2. The authors should add standard deviations. Lines 451-462. Where all these data are reported? The authors should add them as supplementary files. Line 467. “In further experiment cultivation…”. What does it mean? Did the authors performed additional tests? Line 484. Why did the authors introduce the concept of C/N only now? This is a very important characteristic of the medium when lipid production is evaluated. 6. Add the label in all the x axis. Additionally it could be interesting to have lipids, carotenoids, ergosterol,….. production also as % per CDW (also as supplementary material), in order to understand if the obtained values depend or not on the amount of biomass. The authors made these calculations only for glucan; why not for all the considered parameters? 9. Use arrows to point lipid bodies.    Comparison between the obtained production data (g/L or %) and data showed in other papers, using fat media or other media, could be interesting.

Author Response

Response to Reviewer 1

Dear Reviewer,

thank you a lot for detailed review and many usefull recommendations and comment. We try to accept all your notes as follows:

English language was revised in all the text, some formulations were changed and corrected Results and Discussion were shortened and reorganized all minor recommendations were accepted and changes made in the text are highlighted in yellow cultivation experiments were clearly described and some aspects were explained in Methods and Results growth curves of red yeasts were added as Figure 1S to supplementary materials for better explanation of total cultivation time; comments are included to Results and Figure 2 regarding stationary phase Figures 3 and 4 were better described Tables 2, 3 were transferred into supplementary file as Table S2 and Table S1; Tables were renumerated and Schemes removed Table S3 containing summary of all metabolites content in red yeast biomass in percentage per CDW was added to supplementary file

Reviewer 2 Report

In this study, the authors investigate production of various byproducts by red yeasts feeding on animal fat. There is commercial interest in using yeast species to convert animal fat into more refined forms of fatty acids. Understanding the physiology of these yeast species could improve and increase byproduct output. Therefore, this study is of potential interest, but it only adds, at best, moderately to our knowledge of these yeasts. Of concern is the biological replicates that the investigators used. In some cases, the experiments were only repeated twice but the investigators attempt to calculate standard deviation or error. Statistically, it does not make sense to calculate error when you only have two points because the error is truly between point 1 and point 2. Also, in my opinion, several conclusions are wrong, or the data is often over-interpreted. Finally, the paper, as it is written, is overly wordy and repetitive. I found various grammatical and misspelling errors, but I assume that this is due to the fact that the authors are not native English speakers. I did my best to point every writing error in my minor comments section, but I strongly suggest consulting a writing service.  Thus, in my opinion, a thorough a re-write is warranted before publication.

Below I outline point by point major and minor comments.

Major comments.

The authors use the word “strain” when referring to species numerous times. For example, on page 12, line 380 it states “…can be found predominantly in strains R. Mucilaginosa…” This is a species, please refer to it as such. If you are using a strain of this species, please state that, but do not refer to species as strains. This happens numerous times through the article. I noticed that on page 22, lines 603, 604 you do refer to both species and strain number. Please use the same format through the paper. Please define C/N ratio in the abstract or at least in the introduction. Do you mean carbon/nitrogen? I checked the entire document and I cannot find a certain instance where this is defined. The authors mention that some experiments were carried in triplicates and others in duplicates. First, it should be clearly stated which ones are duplicates and which ones are triplicates. Second, you cannot do statistics (error bars) when you only run duplicates. Please, remove the error bars from all the duplicates. For all graphs in Figures, 1,2,3,4 report statistics for the triplicates and graph both lines for the duplicates without statistics. Table 1 should be moved to results. Please format this table as Scheme 1, 2, 3. Scheme 1, 2, 3 should be called Table 1, 2, 3. Then Table 1 from point (4) above should be renamed Table 4 since it is after the “schemes.” The first paragraph of results section is redundant with the intro. Consider shortening, moving to intro, or removing entirely. First paragraph of results section 3.2 is also repetitive. Please remove. Page 9, line 327. Authors mention that stationary phase was not reached. How do you know? A growth curve should be performed before this conclusion is drawn. The next sentence is also confusing. What do the authors mean by compromise? Either provide more evidence or do a growth curve on your studies. This reviewer does not agree with the conclusion drawn on page 11 line 365. For example, lipase production for panel (a), glucose is similar between T=0 and T=96. The readings fluctuate up and down but not much is changed. In fact, for one sample it goes back to 0. It is impossible to conclude from these data whether you just measured resting levels of lipase or you are truly looking at lipase induced due to glucose addition. Similar comments for panels (b) and (d). This is compounded by the fact that some of the samples were duplicates and the statistics are meaningless. The only reasonable conclusions can be drawn from panels (c) and (e). Please move Tables 2 and 3 to supplemental material. Page 13 starting at line 383. The authors give several values, 3X, 7X, etc and refer to table 3. On what do the base this increases? Is there a standard that has been agreed on? Are you comparing T=0 with T=96? Please clarify this. Also, 1.3X increase is meaningless and could be within error. Results section 3.4. Please show the data on emulsification as a supplemental figure. Also, the first paragraph and the first sentence in the second paragraph are redundant. You are restating methods. Table 2. As I stated before, the average values and most importantly error or standard deviation of 2 measurements are generally meaningless when doing statistics. Please move this table to supplemental and list both values for each measurement. Page 15, lines 460-466. This reviewer does not agree with the conclusions in this paragraph. Considering that only two measurements are done, a 1.3X increase is nearly impossible to conclude. Increases of 3-7X should be discussed but in my opinion no conclusion on increase production can be drawn for the other measurements. Page 17, line 520. Again, is hard to conclude 1.5-2X increase when only two measurements were done. Please move Fig.8 to supplemental. You cannot interpret what you are looking at. You don’t know for sure that what you are seeing are polysaccharides. Figure 9 is informative, and the conclusions are justified. The discussion section should be significantly shortened. I have given up reading it because most of the time you are restating either introduction or methods. I do like the “conclusion” section. I think that adding one or two more paragraphs to that should suffice. As it stands now, the discussion is too tedious and not very informative. For example, in lines 650-697 you are restating introduction and is tedious to re-read. Begin with paragraph on page 698 and briefly state your previous results within one or two sentences. Then discuss the data in this paper. Importantly, discuss the significance of your experiments in the light that some of them were only repeated twice.

Minor comments.

Genus and species should be italicized. See abstract, line 19. Last line of abstract is awkward. Please rephrase. Page 1, line 36. “that is fit for human consumption.” Lines 39-41 appear to contradict previous sentence. Is this fat intended for human consumption or not? Please clarify. Line 45. What do you mean by “for value creation.”? Please clarify Line 46. Extent not extend. Line 51. Please use the word “isomerizing” rather than “re-arranging”. Line 54. What do you mean by “upgradation.”? Line 55. Please say: “This process does not require expensive equipment…” Lines 74, 75. “Wide range” used twice with two different meanings. Consider using a synonym. Line 93. “Except for our previous…” and then “to our best knowledge…” Line 94: “first detail report on use of crude…” I assume you don’t mean rude! Line 98. Please define GRAS and PUFA. I don’t see a definition earlier for either. Line 105. “It is worth noting.” Line 122. “Based on our preliminary…” Line 130. What do you mean by “revealed.” May want to use another word. Table 1 legend, line 140. Lipases not lipase. Line 145. Use “consistently” rather than “permanently.” Line 153. What do you mean by volume ration v/v? Please define. Line 173. “Cultivation was finished after 96hrs” Please remove this sentence. Is redundant. Line 197. “Supernatant”. What does this word mean? Did you spin down the yeast? Or did you just analyze some of the yeast culture. Line 209. What does “orientational” mean? Preliminary? Line 210. Cells were cultivated “on” glucose. Also “To induction of biosurfactant…” is grammatically incorrect. Please restate. Line 214. First sentence of this paragraph is awkward. Please rephrase. Line 223. Please add space between 5 and times. Line 230. “Mobile phase B was composed of methanol…” After reformatting the Schemes as Tables, please refer to them as Tables, 1, 2, 3 in the text. Line 280. Delete also from the sentence. Section 2.6 lines 283, 284. Please state which experiments were duplicate and which were triplicate. The choice of colors for the lines in the figures should be improved. The different shades of grey are hard to read. I suggest making each line a different color. Figure 2. Please label the Y axis better: “g/L biomass”. Change the title to panel A as “glucose” to match the other titles. Ensure that the fonts for the Y and X-axis labels are the same. Figure 3. Please label all the panels as in Figure 1: glucose, glycerol, crude fat, etc. Move the pH label from panel A next to the Y axis. Figure 4. Same comments as figures 2 and 3. Additionally, Fig.4 has no Y axis or X-axis labels. The (a), (b), (c), etc labels are also placed at different positions under the panels and of different fonts. Please ensure that everything is consistent. Page 13, line 395. Sentence starting with “Based on data measured…” is awkward. Please rephrase or split into two sentences. Page 13, line 422. You are referring to table 4 before table 3 and as far as I can tell also before Table 2. Please re-label these tables to be sequential. Also move these tables (3 and 4) to supplemental material. Table 2 should stay in the main paper because is data that has not been discussed in any other figure. Figure 5. For X-axis please use “species” not “strain.” Move the label “A-index” on the Y-axis not inside the panel. Results section 3.5.1. First paragraph redundant with materials and methods. Remove. Page 15, lines 467-470. Redundant with materials and methods. Page 16, line 484. What do you mean by “13 red yeasts”? You mean 13 strains representing 4 species? Page 17, first paragraph of section 3.5.2. Redundant with materials and methods. Figure 6. Carefully label the Y-axis for all panels. Also, different fonts are used for the different panels. Compare (c) lipids with (d) carotenoids. Further, the X-axis of each figure is misleading. The numbers 13, 25, 50, 100 are present under some bars but not others. Please be consistent. Also, please be careful with the design of these panels. Some are bigger, some are smaller. Some are placed in the middle of the square, some are to the side, etc. Page 17, line 505. What do the authors mean by “fluctuated values with growing trend”? Similar comments on Fig.7 as previous figures. Be careful with the labels. Also, what time were these measurements taken at? I cannot tell. I commend the authors for not re-stating the methods in results section 3.5.3. But please indicate the time in methods section. Page 22, line 608. The first sentence in that paragraph does not make sense. Please re-state. Line 650. You may not want to use the word “perfect.” Nothing is truly perfect. May want to use “great” instead. Line 653. Unnecessary indentation. Line 714. Similarly to the screen study, the amount… Lane 719. Sentence redundant with lane 714. Please remove and place reference 19 after the sentence starting on line 714. Line 725. What do you mean by “taken slower”? Lane 727. A recent study [34], has shown that… Lane 729. The not tne. Line 741. Thus, in the present study… Line 742. Delete was. Line 755. Despite the fact that lipases…

Author Response

Response to Reviewer 2 comments:

Dear Reviewer,

thank you for detailed revision of our paper and for many recommendations and proposed corrections. There are many comments to our manuscript and major revision was done. We also tried to correct all mistakes. In this response main revisions are summarized; corrections in sctions Introduction and Methods were highlighted in yellow directly in the text. Sections Results, Discussion and Figures + Tables were changed substantially and rewritten.

English language was substantially revised in all the text, some parts were rewritten number of repetitions of cultivation experiments were clearly explained in Methods and added to comments in Results and Figures/Tables - most of coltivations was done in triplicates and, thus, generall interpretation of data is valid results were more clearly interpreted and discussion was rewritten and substantially shortened; nevertheless, because of amount and complexity of results some discussion is still needed strain numbers were added to each yeast strain in all the text C/N ratio was defined in abstract we did not clearly understand recommendations regarding Tables and Schemes numbers; manuscript was corrected according to recommendation of Reviewer 1 - we hope that correctly Table 1 was renamed as Table 4; Tables 2 and 3 were moved to supplementary material (as Table S2 and Table S1); Table 4 was renamed as Table 5; Table S3 was added to supplementary material as recommended Rev.1 all figures were corrected according to recommendations (colors, titles, labeling, axes description, fonts etc.); Figure 8 was moved to Supplemental; Figure S1 containing growth curves was added for better explanation of cultivation time Figure S2 containing results of oil spreading test was added into Supplemental Table 5 containing data of glucans stayed in the main text (these data are not discussed in any other figure)

All minor comments were accepted in Introduction, Methods and Results, Tha section Discussion was rewritten and, some parts were removed, thus, some proposed corrections were not included.

Round 2

Reviewer 2 Report

The authors have made significant changes to this paper. This reviewer is now satisfied with the presentation and interpretation of the results. In my opinion this paper is now suited for publication. 

I only have minor comments on formatting of some figures. For example, in figure 3 panel (e) there seems to be a space after e before the parenthesis close: (e ) not (e).

Some labels on the other figures also require some formatting. 

However, I assume that the copy editor will catch these minor errors. 

My recommendation is to accept after minor editing revisions. I do not detect any methodology errors.